# Improvement of immune dysregulation in individuals with long COVID at 24-months following SARS-CoV-2 infection

Chansavath Phetsouphanh [1] ✉, Brendan Jacka[1], Sara Ballouz[2,3], Katherine J. L. Jackson [2], Daniel B. Wilson[1], Bikash Manandhar[1], Vera Klemm[1], Hyon-Xhi Tan [4], Adam Wheatley[4], Anupriya Aggarwal[1], Anouschka Akerman[1], Vanessa Milogiannakis[1], Mitchell Starr[5], Phillip Cunningham[5], Stuart G. Turville [1], Stephen J. Kent [4], Anthony Byrne[6], Bruce J. Brew[7], David R. Darley [8], Gregory J. Dore[1,8], Anthony D. Kelleher [1,8,9] ✉ & Gail V. Matthews[1,8,9] ✉

This study investigates the humoral and cellular immune responses and health-related quality of life measures in individuals with mild to moderate long COVID (LC) compared to age and gender matched recovered COVID-19 controls (MC) over 24 months. LC participants show elevated nucleocapsid IgG levels at 3 months, and higher neutralizing capacity up to 8 months post-infection. Increased spike-specific and nucleocapsid-specific CD4[+] T cells, PD-1, and TIM-3 expression on CD4[+] and CD8[+] T cells were observed at 3 and 8 months, but these differences do not persist at 24 months. Some LC participants had detectable IFN-γ and IFN-β, that was attributed to reinfection and antigen re-exposure. Single-cell RNA sequencing at the 24 month timepoint shows similar immune cell proportions and reconstitution of naïve T and B cell subsets in LC and MC. No significant differences in exhaustion scores or antigen-specific T cell clones are observed. These findings suggest resolution of immune activation in LC and return to comparable immune responses between LC and MC over time. Improvement in self-reported health-related quality of life at 24 months was also evident in the majority of LC (62%). PTX3, CRP levels and platelet count are associated with improvements in health-related quality of life.

Three years after the World Health Organization (WHO) declared a pandemic, and with >600 million cases globally, the burden of disease attributable to post-acute COVID-19 is a major public health issue. While the vast majority of people now survive acute infection, significant morbidity may persist for months following acute infection[1]. One manifestation of this is the phenomenon known colloquially as 'long COVID'[2]. Although there is no single accepted definition, this condition generally encompasses various physical and

[1]The Kirby Institute, University of New South Wales, Sydney, NSW, Australia. [2]Garvan Institute for Medical research, Sydney, NSW, Australia. [3]School of Computer Science and Engineering, Faculty of Engineering, University of New South Wales, Sydney, NSW, Australia. [4]Department of Microbiology and Immunology, Peter Doherty Institute, University of Melbourne, Victoria, VIC, Australia. [5]NSW State Reference Laboratory for HIV, St. Vincent's Centre for Applied Medical Research, Sydney, NSW, Australia. [6]Heart Lung Clinic, St. Vincent's Hospital Sydney and Faculty of Medicine and Health (UNSW), Sydney, NSW, Australia. [7]Peter Duncan Neurosciences Unit- St Vincent's Centre for Applied Medical Research, Sydney, NSW, Australia. [8]St. Vincent's Hospital, Darlinghurst, NSW, Australia. [9]These authors contributed equally: Anthony D. Kelleher, Gail V. Matthews. ✉e-mail: cphetsouphanh@kirby.unsw.edu.au; akelleher@kirby.unsw.edu.au; gmatthews@kirby.unsw.edu.au

neuropsychiatric symptoms lasting longer than 12 weeks[3–5]. 'Long COVID[1]'or post-acute sequelae of COVID-19 (PASC) (henceforth LC) significantly contributes to COVID-19 related morbidity, initially complicating the long-term management of a large proportion of hospitalized patients. Amongst community managed COVID-19 cases, prevalence of persisting symptoms is lower, but remains higher than would be expected given the often-mild nature of the acute illness. Consistent with international data, initial reports from the first wave of infection in Australia suggested rates of LC between 10 and 30% of community managed unvaccinated individuals at 2–3 months post infection[6,7], with symptoms persisting up to 8 months[8]. In vaccinated patients, and with evolving variants including Omicron, estimates of LC prevalence are lower, generally less than 5%[9–12]. However, given the massive global burden of infection this equates to millions of potential LC cases.

Defining rational and evidence-based therapies for this complex condition is dependent on understanding its pathogenesis. Previously in a matched control design we revealed a distinct immunological footprint in those with LC compared to recovered individuals[8]. This footprint is characterized by long-lasting (>8 months) inflammation, of certain innate immune (monocytes and plasmacytoid dendritic) cells, activation of subsets of CD8$^+$ memory T cells (expressing PD-1 and Tim-3), and the sustained production of two specific antiviral cytokines (IFN-β and IFN-λ1). Importantly, these seminal observations have recently been independently confirmed by other groups[13,14]. While this demonstrates that LC is associated with a distinctive pattern of immune dysregulation, it does not tell us the drivers of this dysregulation. Four possible mechanisms have been proposed by us[8] and others[15–17]: (1) Persistence of SARS-CoV-2 antigenic material, (2) an autoimmune phenomenon, (3) repeated cycles of damage and repair in tissues, or (4) other mechanisms.

In addition to uncertainty regarding pathogenesis, major questions persist regarding long-term trajectory of LC symptoms and the degree of recovery over time experienced by individuals. Impacts to the patient's quality of life, capacity to return to work, and onus on healthcare systems are significant and critically dependent on patterns of return to health. Furthermore, little is currently known about how immuno-pathological measures correlate with improvements in quality of life. Here, we show the temporal trends in immunological and pathological biomarkers and self-reported quality of life up to 24 months after acute infection, in patients with mild/moderate SARS-CoV-2 infection from ancestral strain. Participants with LC, compared with asymptomatic aged and gender matched controls who have recovered from COVID-19 (MC), are comprehensively assessed regarding immune phenotypes and T cell function within a longitudinally followed cohort (ADAPT) up to 24 months post-infection. Complex clinical scores pertaining to quality of life at 24 months are prospectively evaluated and both clinical and laboratory datasets modeled to ascertain immune parameters associated with recovery.

## Results
### Cohort characteristics
The ADAPT cohort enrolled individuals with confirmed COVID-19 from mid-2020, with around 90% of cases community-managed. From this cohort (n = 62), sub-groups of participants with LC (occurrence of one of three major symptoms; fatigue, dyspnea, or chest pain) defined at 4-months (median of 128 days) (n = 31) and age and gender matched asymptomatic convalescent controls (MC)(n = 31) were followed for 2-years with detailed immunological and clinical evaluation (Supplementary Fig. 1). At 24-months 23% (n = 7) of LC and 26% (n = 8) of MC were lost to follow-up (Table 1). Of note, all participants in this sub-study were unvaccinated at acute infection and at enrolment, but most (85%) were subsequently fully vaccinated between 12- and 24-month timepoints [median 474 days (IQR: 429, 507) to first vaccination], with no difference between the groups in terms of vaccine type. A small

**Table 1 | Demographic and clinical characteristics of participants in the ADAPT Study**

|  | Total | Matched Control | Long-COVID |
|---|---|---|---|
|  | N = 62 [n (%)] | N = 31 [n (%)] | N = 31 [n (%)] |
| Age (median, IQR) | 50.5 (40–60) | 50 (39–60) | 51 (40–60) |
| Gender |  |  |  |
| Female | 32 (52%) | 16 (52%) | 16 (52%) |
| Male | 30 (48%) | 15 (48%) | 15 (48%) |
| Race/ethnicity |  |  |  |
| Caucasian/White | 54 (87%) | 26 (84%) | 28 (90%) |
| Other than Caucasian/White | 8 (13%) | 5 (16%) | 3 (10%) |
| Enrolment source |  |  |  |
| Community | 52 (84%) | 29 (94%) | 23 (74%) |
| Inpatient | 10 (16%) | 2 (6%) | 8 (26%) |
| Complete 24-month visit |  |  |  |
| No | 15 (24%) | 8 (26%) | 7 (23%) |
| Yes | 47 (76%) | 23 (74%) | 24 (77%) |
| Vaccine type |  |  |  |
| No vaccination | 9 (15%) | 5 (16%) | 4 (13%) |
| AstraZeneca | 27 (44%) | 13 (42%) | 14 (45%) |
| Pfizer | 26 (42%) | 13 (42%) | 13 (42%) |
| Days to first vaccination (median, IQR)[a] | 474 (429, 507) | 480 (429, 517) | 467 (426, 490) |
| New COVID infection during follow-up |  |  |  |
| No | 46 (74%) | 23 (74%) | 23 (74%) |
| Yes | 16 (26%) | 8 (26%) | 8 (26%) |
| Routine pathology at 4-months post infection |  |  |  |
| C reactive protein | 0.8 (0.4–1.6) | 0.8 (0.4–2.0) | 0.9 (0.5–1.5) |
| D-Dimer | 0.3 (0.3–0.4) | 0.3 (0.3–0.3) | 0.3 (0.3–0.4) |
| Lymphocyte count | 1.6 (1.4–2.0) | 1.5 (1.4–1.8) | 1.8 (1.4–2.2) |
| Total cholesterol | 5.0 (4.3–5.7) | 4.8 (4.3–6.2) | 5.0 (4.3–5.6) |
| Troponin I | 2.0 (2.0–4.0) | 2.0 (2.0–4.0) | 3.0 (2.0–4.0) |
| BSL /Glucose | 5.0 (4.6–5.4) | 4.8 (4.7–5.4) | 5.0 (4.6–5.3) |
| Neutrophil count | 3.0 (2.4–3.6) | 3.0 (2.4–4.0) | 2.9 (2.4–3.5) |
| Platelet | 223 (196–251) | 211 (194–252) | 227 (201–251) |

[a]Among those receiving vaccination.

proportion of participants were reinfected prior to the 2-year visit (LC n = 8 of 24 and MC n = 8 of 23, [median 673 (IQR: 655, 718) days since initial infection]). Routine pathology assays including C reactive protein (CRP), D-Dimer, total cholesterol, platelet count, troponin I, glucose, lymphocyte, and neutrophil counts were measured with no differences observed between LC and MC for any time point (4-, 8- or 24-months).

### Elevated neutralizing antibodies in long COVID prior to vaccination
To evaluate humoral response following infection with ancestral SARS-CoV-2, serum antibody levels and cellular components were measured. Total spike IgG levels were elevated in LC participants compared to MC, with an average of 2.1-fold higher IgG in serum between 3- and 12-months, albeit not significantly higher (Fig. 1A). No difference in spike IgG levels were observed at 24-months. IgG antibodies directed against nucleocapsid protein (NP) were 3-fold higher in LC at 3-months compared to MC (median [IQR]: 5.01 [1.41 and 5.89] versus 1.67 [1.02 and 4.43]; p = 0.035) (Fig. 1B). This trend continued up to 12-months, with an average of 2.6-fold higher anti-NP IgG levels in LC. No difference was observed at 24-months. Neutralizing capacity of anti-spike IgG within the two groups was evaluated by utilizing a live virus neutralization

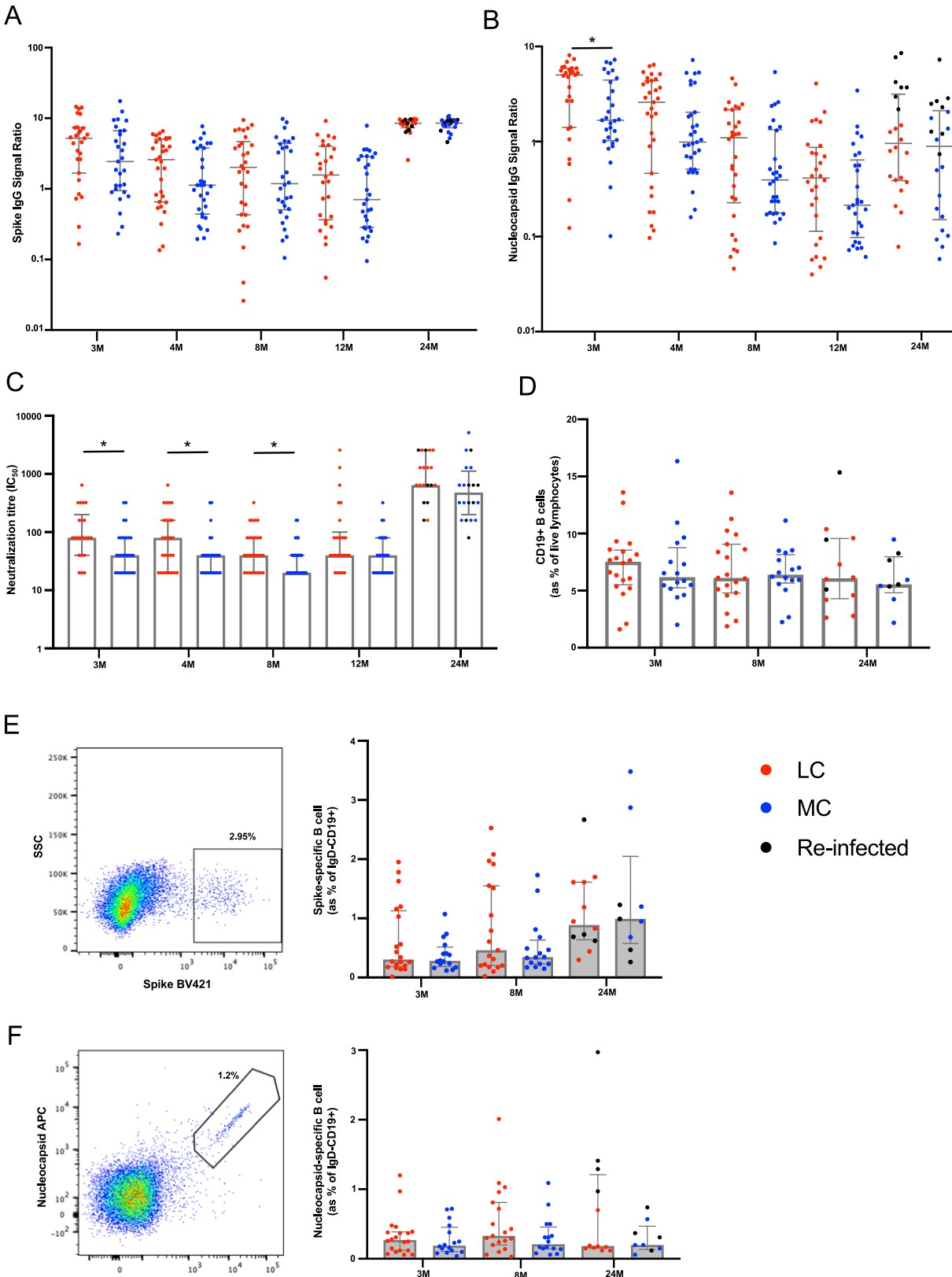

**Fig. 1 | Humoral response in participants with long COVID following SARS-CoV-2 infection. A** Anti-spike IgG levels were elevated in LC at earlier timepoints. **B** Higher anti-nucleocapsid were significantly higher at 3-months in LC. **C** Neutralization tired remained higher in LC up to 8-months post-infection. There was no difference following vaccination at 12- or 24-months. **D** Similar frequencies of bulk CD19 + B-cells between the two groups. **E** Representative dot plots showing spike tetramer binding from memory (IgD-) B cells. Increased frequencies of spike-specific B cells evident following vaccination. **F** Representative dot plots showing nucleocapsid tetramer binding from memory (IgD-) B cells. Data shown as medians with interquartile ranges. LC= Long COVID, MC= matched controls, M= months. Mann–Whitney U test (two-sided) was used for un-paired analysis, $p < 0.05$ (*) were considered significant. Data points represent $n = 31$ biologically independent samples per group. Source data are provided as a Source Data file.

assay. Half of maximal neutralization ($IC_{50}$) levels were, on average, 2-fold higher in LC compared to MC at 3-, 4- and 8-months ($p = 0.014$, $p = 0.045$, $p = 0.038$; respectively) (Fig. 1C). At post-vaccination time-points (12- and 24-months), neutralization titres were similar. There were no discernible disparities between bulk CD19$^+$ B cell frequencies (Fig. 1D), spike tetramer-specific memory B cells (Fig. 1E), or nucleocapsid-specific memory B cells (Fig. 1F) at either 3-, 8- or 24-months within LC and MC.

## Sustained CD8 T responses to SARS-CoV-2 antigens up to 24-months with comparable levels of PD-1 and TIM-3 expression

Activation induced markers and inhibitory check-point receptor expression were used to assess T cell profiles (Supplementary Fig. 2). To assess recall memory[18,19], surface co-expression of CD25 and CD134 was measured on CD4$^+$ T cells at 48 h following stimulation with SARS-CoV-2 peptides. Spike-specific CD4$^+$ T cells were 3.9-fold higher in LC at 3-months ($p = 0.014$). This was maintained at 8-months (2.2-fold; $p = 0.014$) but decreased at 24-months (1.5-fold; $p = 0.149$) (Fig. 2A). Similarly, CD4$^+$ T cell responses toward nucleocapsid peptides were elevated in LC at 3- (1.5-fold, $p = 0.030$) and 8- months (4.6-fold, $p = 0.007$), however by 24-months this finding was no longer observed (1.2-fold, $p = 0.268$) (Fig. 2B). SARS-CoV-2 reactive CD8$^+$ T cells were measured using the co-expression of CD69 and CD137. A 2.8-fold higher frequency of spike-specific CD8$^+$ T cells were found in LC at 3-months ($p = 0.031$) that increased at 8-months (5.6-fold; $p = 0.007$) and was maintained at 24-months (2.7-fold; $p = 0.004$) (Fig. 2C). Similarly, nucleocapsid-specific CD8$^+$ T cell responses were elevated in LC at 3- (4.2-fold, $p = 0.029$), 8- (5.1-fold, $p = 0.029$), and 24- months (1.7-fold, $p = 0.034$) (Fig. 2D).

Surface expression of inhibitory receptor PD-1 on bulk CD4$^+$ T cells did not differ between LC and MC at any timepoint (Fig. 2E). Higher levels of PD-1 were observed on CD8$^+$ T cells at 3- (1.7-fold, $p = 0.039$) and 8-months (1.5-fold, $p = 0.023$) (Fig. 2F), however these were similar at 24-months. T cell inhibitory marker TIM-3 was higher in LC at 3- (1.3-fold, $p = 0.041$) and 8-months (1.5-fold, $p = 0.031$) on CD4$^+$ T cells (Fig. 2G) and at 3-months (1.8-fold, $p = 0.048$) in the CD8 T cell subset (Fig. 2H). However, levels of TIM-3 showed no difference at 24-months.

## Reduced innate immune cell activation but detectable levels of IFN-β

Six serum analytes (IL-6, PTX3, IFN-l1, IFN-γ, IFN-λ2/3 and IFN-β) that were highly associated with LC in our previously established log-linear model[8] were also measured at 24-months. IFN-γ remained detectable in some LC participants compared to MC (1.9-fold, $p = 0.021$). In addition, IFN-β was also more elevated in LC (1.5-fold, $p = 0.010$) (Fig. 3A). Cellular activation (as measured by co-expression of HLA-DR and CD38) of monocytes was evident at 3- (2.1-fold, $p = 0.037$) and 8-months (4.02-fold, $p = 0.0004$) but resolved at 24-months (Fig. 3B). Frequencies of activated myeloid dendritic cells (mDC) were comparable between LC and MC at all timepoints. Importantly, a reduction of activated mDCs were evident in both groups at 24-months compared to 3-months (5.5-fold decrease in LC and 8.8-fold in MC) (Fig. 3C). Like monocytes, activated plasmacytoid dendritic cells (pDC) were increased in LC at 3-months (1.6-fold, $p = 0.022$) and 8-months (1.4-fold, $p = 0.029$), and a reduction of activation was observed in both groups at 24-months (Fig. 3D).

## Reconstitution of immune subsets at 2-years post-infection

To elucidate the cellular immune profile longitudinally, we utilized 10X genomics 5′ single-cell RNA sequencing platform. Whole transcriptome coupled with T cell receptor (TCR) and B cell receptor (BCR) sequence analysis was performed on $n = 10$ LC and $n = 10$ MC, with a total of 79,006 cells passing stringent QC and filtering. 31 clusters denoting differing immune subsets were identified following dimensional reduction of sequence data employing Uniform Manifold Approximation and Projection (UMAP). When comparing between groups LC had lower levels of naïve CD4$^+$ T cells (2.1-fold less, $p = 0.02$) and naïve CD8$^+$ T cells (1.2-fold less, $p = 0.01$) compared to MC at 8-months. There was no difference in innate cell subsets between the two groups at any time-point. In order to study cellular profiles in each group, the time courses of changes in subsets were tracked from 4 months over time. This showed: increases in CD14$^+$ monocytes (1.8-fold, $p = 0.01$) and memory B cells (2.9-fold, $p = 0.007$) in MC by 24-months (Fig. 4A). In LC, naïve B cells (2.0-fold, $p = 0.009$), memory B cells (1.9-fold, $p = 0.04$) and naïve CD4$^+$ T cells (2.1-fold, $p = 0.04$) increased over time to 24 months. There were no clear differences between immune cell profiles between LC and MC at 24-months as shown in the UMAP (Fig. 4B). Frequencies of immune cells were similar in both innate and adaptive compartments. Naïve T cell subsets were slightly lower in LC but not significantly so (CD4 naïve, LC = 2.15% versus MC = 2.66%, $p = 0.74$; CD8 naïve, LC = 4.11% versus MC = 6.58%, $p = 0.33$) (Fig. 4C). Additionally, dimensional reduction of naïve B and T cell subsets showed lack of divergence between the LC and MC groups (Fig. 4D), suggesting reconstitution of cells within these subsets over time.

Given our initial finding of increased levels of interferon-β and interferon-λ1 at 4 and 8 months[8], interferon response scores (IRS) were calculated using differentially expressed genes downstream of inter-feron signaling pathway encompassing ~360 genes[20,21] for the LC and MC groups. Comparable IRS were observed in most innate cell subsets besides CD14$^+$ monocytes, where IRS was higher in LC at 24-months (mean weighted score [MWS]; LC = 3.92 versus MC = 3.46, adjusted $p = 1.22E^{-16}$) (Fig. 4E). IRS in Natural Killer (NK) cells were higher in LC compared to MC at 4-months (MWS; LC = 2.57 versus MC = 2.34, adjusted $p = 3.58E^{-5}$) but no differences were observed in the later timepoints. CD4$^+$ T cells from LC had higher IRS at 4-(MWS; LC = 1.61 versus MC = 1.22, adjusted $p = 3.98E^{-28}$) and 24-months (MWS; LC = 0.36 versus MC = 0.28, adjusted $p = 2.33E^{-4}$). While CD8$^+$ T cells in LCs had higher IRS throughout: 4-months (MWS; LC = 1.76 versus MC = 1.27, adjusted $p = 3.95E^{-29}$), 8-months (MWS; LC = 1.20 versus MC = 1.15, adjusted $p = 4.63E^{-2}$) and 24-months (MWS; LC = 0.37 versus MC = 0.24, adjusted $p = 2.29E^{-11}$). Importantly, it should be noted that the IRS MWS decreased in LCs at 24-months from 4-months by 4.4-fold for CD4$^+$ T cells and 4.7-fold for CD8$^+$ T cells.

T cell exhaustion scores[22] calculated from ~282 exhaustion/inhibition related genes were examined for 11 T cell subsets and no difference was observed between LC versus MC for any of the subsets analyzed (Fig. 4F). Furthermore, antigen-specificity of T cell clones with paired α and β chains were referenced to immuneCODE and VDJdb databases to ascertain TCR specificity. SARS-CoV-2, CMV, EBV, influenza, 'self' Homosapien reactive clones and clones that mapped to multiple anti-gens were evident in both groups (Supplementary Fig. 3A). One LC donor had expanded 'self' reactive T cell clones directed to Insulin-like Growth Factor 2 mRNA-binding protein 2 (*IGF2BP2*) that was associated with their being a type II diabetic. No discernible differences were observed in the exhaustion state of any antigen-specific clones including SARS-COV-2 specific T cells (Supplementary Fig. 3B).

## Recovery of health-related quality of life at 2-years post-infection

Self-reported health-related quality of life was assessed through the validated EQ-5D-5L index score collected at all timepoints. Participants with LC more often reported problems with mobility, usual activities, and pain/discomfort EQ-5D-5L domains at 4-month visit, but by 24-months no significant differences were observed between the groups (Table 2). Participants with LC had a significantly lower EQ-5D index score 4-months [0.87 (IQR: 0.80, 0.94)] compared to matched controls [0.94 (IQR: 0.92, 1.00); $p$ value: 0.001], however there was no sig-nificant difference in median EQ-5D-5L index scores at 24-months by LC status 0.92 (IQR: 0.83, 0.93) and 0.93 (IQR: 0.86, 1.00) for LC and MC, respectively; $p$ value: 0.16 (Fig. 5A–C).

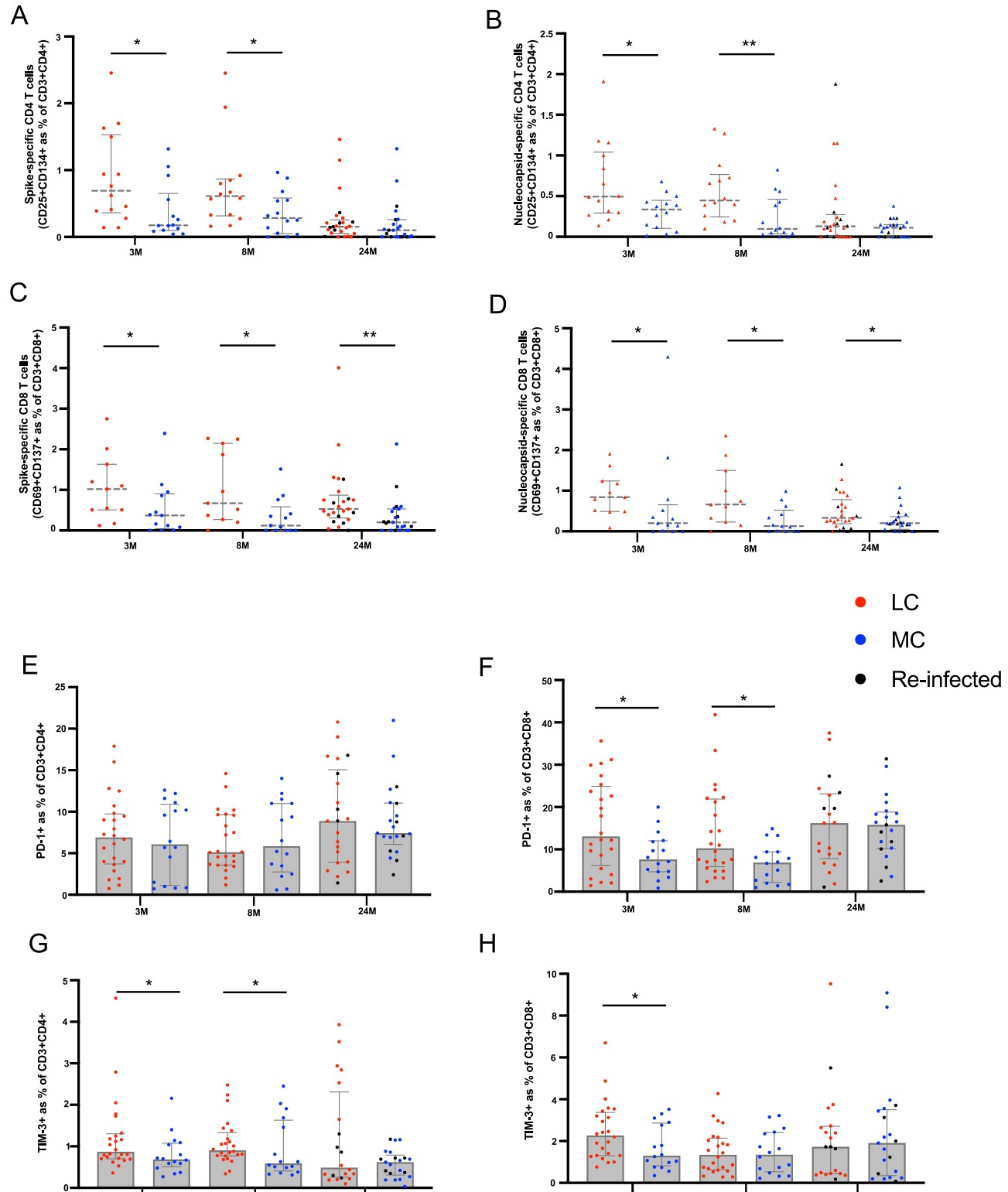

**Fig. 2 | T cells responses and Inhibitory marker expression in participants with long COVID. A**, **B** Spike-specific and nucleocapsid-specific CD4+T cell responses were higher in LC at 3- and 8-months. No difference at 24-months. **C**, **D** Sustained spike-specific and nucleocapsid-specific CD8 + T cell responses across all 3 time-points. **E** No difference in PD-1 expression on CD4+ T cells. **F** Elevated PD-1 expression on CD8+ T cells at 3- and 8-months. **G**, **H** Increased TIM-3 expression on CD4 and CD8 T cells at earlier timepoints. Data shown as medians with interquartile ranges. LC= Long COVID, MC= matched controls. Mann–Whitney U test (two-sided) was used for un-paired analysis, $p < 0.05$ (*), $< 0.01$ (**) were considered significant. Data points represent $n = 31$ biologically independent samples per group. Source data are provided as a Source Data file.

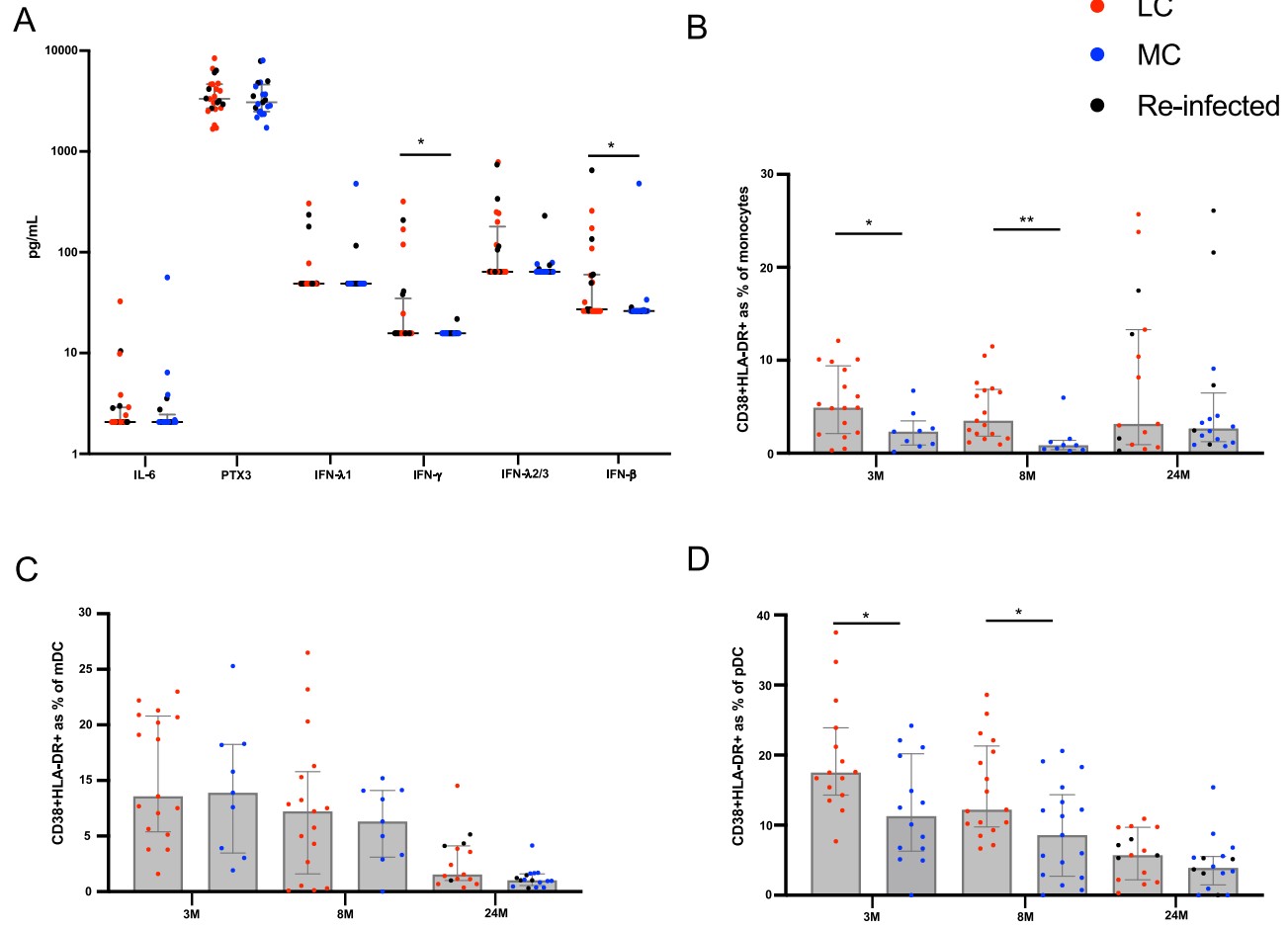

**Fig. 3 | Reduction of immune activation at 24-months. A** Significantly elevated IFN-γ and IFN-β in LC at 24-months. **B** Higher frequencies of activated monocytes at 3- and 8-month, but not 24-months. **C** Lower percentages of activated myeloid dendritic cells (mDC) at 24-months. **D** Higher percentages of activated plasmacy-toid dendritic cells (pDC) at 3- and 8-month that decreased 24-months. Data shown as medians with interquartile ranges. LC Long COVID, MC matched controls. Mann–Whitney U test (two-sided) was used for un-paired analysis, $p < 0.05$ (*), $< 0.01$ (**) were considered significant. Data points represent $n = 31$ biologically independent samples per group. Source data are provided as a Source Data file.

At 4-months when compared to sex- and age-matched population with normative health values in Australia, participants with LC were more likely to have "poor health" compared to MC: 58% of LC participants had an EQ-5D-5L index score below the lower 95% confidence interval of population normative values ("poor health") at this timepoint versus 16% for matched controls ($p = 0.001$). By the 24-month visit, there was no statistical difference in the proportion of participants with "poor health" when stratified by LC status although the proportion with poor health remained higher in the LC group (38% vs 26% for LC and MC, respectively; $p = 0.53$). To elucidate participant perceptions of their own recovery, participants were asked for their agreement with four statements about the impact of COVID-19 on daily functioning at 8-, 12- and 24-month visits (Supplementary Fig. 4). Participants with LC at 24-months were less likely to agree with statements about making a full recovery from COVID-19 (42% vs. 5% for LC and MC, respectively; $p = 0.004$), return to usual activities (17% vs. 0% for LC and MC, respectively; $p$ value: 0.017), and return to exercise (38% vs. 5% for LC and M, respectively; $p$ value: 0.015). There was no difference in return to pre-COVID work by LC status at 24-months (Fig. 5D).

**Blood markers associated with improvement of health-related quality of life**

An established log-linear classification model[8] was used to analyze 15 blood parameters (IL-6, PTX3, IFN-λ1, IFN-γ, IFN-λ2/3, IFN-β, CRP, D-dimer, platelets, troponin, cholesterol, blood sugar level, neutrophils,

lymphocyte count and neutrophil: lymphocyte ratio) from LC participants at 24-months and associations with improvement of health-related quality of life were ascertained. The most prominent features that were associated with improvement of health-related quality of life were PTX3, CRP and platelet levels (Fig. 6A). The top 2 features being PTX3 and platelet count giving an accuracy of 71% and F1 score of 0.78. By adding CRP, accuracy increased to 73% with an F1 score of 0.80 (Fig. 6B). Levels of these 3 analytes were stratified between participants in LC group into recovered (improvement in health-related quality of life) and unrecovered (no improvement in health-related quality of life) and then compared to MC group. Beyond identifying the optimal set of blood markers that are most highly associated with recovery, log-linear classifiers define what is known as a decision boundary. A participant's concentration of the 3 aforementioned markers at 24-months will lie on either side of this boundary, and its positioning relative to the boundary will determine the association between recovered or unrecovered. The decision boundary for PTX3, Platelets and CRP are three-dimensional (Fig. 6C, left panel) and the domain boundary can be clearly visualized with two-dimensional projections (Fig. 6C, right panels).

## Discussion

Our study comprehensively evaluates immunological and clinical parameters in individuals who contracted SARS-CoV-2 infection during the first year of the pandemic, prior to the availability of vaccines or

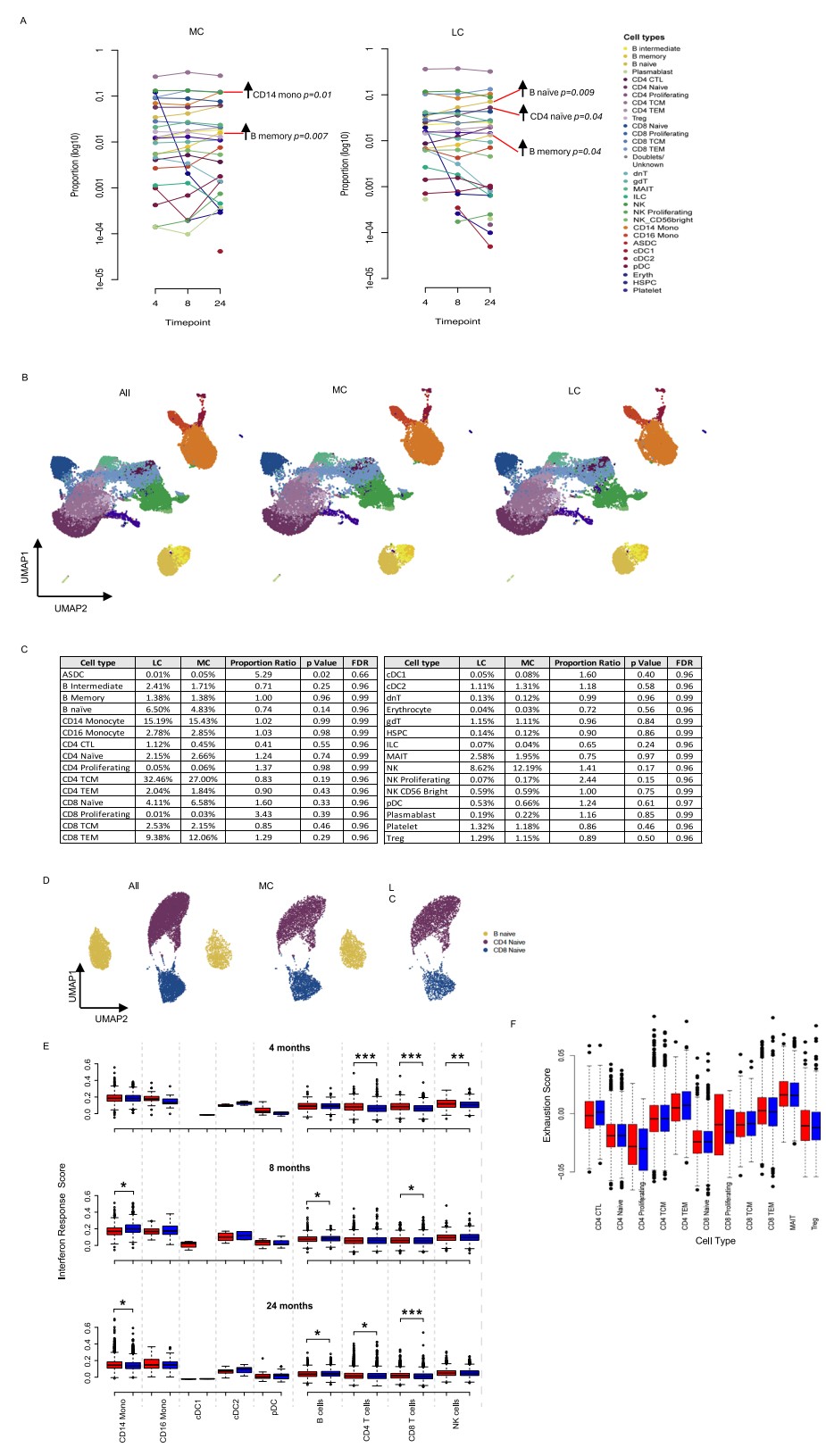

**Fig. 4 | Reconstitution of immune cell subsets at 24-months. A** Longitudinal plot showing changes in cell subset proportions overtime (3-, 8- and 24-months) in MC (left) and LC (right) (**B**) UMAP showing cellular composition single-cell RNAseq data; all (combined cells [45,988 cells]) then separated into LC (*n* = 10) and MC (*n* = 10) at 24-months. **C** Composition of cell subset frequencies between LC and MC with *p* values and false discovery rate (FDR). **D** UMAP assessing only naïve B and T cell subsets, with no clear difference between LC or MC. **E** Box and whiskers graph (median with IQR) showing interferon response scores (IRS) in innate cell subsets. Elevated IRS in CD14+ monocytes. **F** Exhaustion score in T cell subsets. Dots represent individual cells (outliers) for each subset with weighted scores (median with IQR). LC Long COVID, MC matched controls. Data shown as mean scores. *p* < 0.05 (*), < 0.0001(****) were considered significant. Data represent *n* = 10 biologically independent samples per group. Source data are provided as a Source Data file.

**Table 2 | Health-related quality of life scores**

| Measure | 4-month visit [n (%)] | | P value | 24-month visit [n (%)] | | P value |
|---|---|---|---|---|---|---|
| | Matched Control (n = 31) | Long COVID (n = 31) | | Matched Control (n = 23) | Long COVID (n = 24) | |
| Report any problems: | | | | | | |
| Mobility | 0 (0%) | 6 (19%) | 0.024 | 3 (14%) | 5 (21%) | 0.70 |
| Personal care | 0 (0%) | 1 (3%) | 1.00 | 1 (5%) | 1 (4%) | 1.00 |
| Usual activities | 2 (6%) | 15 (48%) | <0.001 | 2 (9%) | 6 (25%) | 0.25 |
| Pain/discomfort | 7 (23%) | 17 (55%) | 0.018 | 11 (50%) | 11 (46%) | 1.00 |
| Anxiety/depression | 11 (37%) | 19 (61%) | 0.074 | 9 (41%) | 17 (71%) | 0.073 |
| EQ-5D-5L index [median (IQR)] | 0.94 (0.92–1.00) | 0.87 (0.80–0.94) | 0.001 | 0.93 (0.86–1.00) | 0.92 (0.83–0.93) | 0.16 |
| "Poor health" status[a] | 5 (16%) | 18 (58%) | 0.001 | 6 (26%) | 9 (38%) | 0.53 |

[a]Defined as ED-5D-5L value below the age- and sex-matched lower 95% CI of normative population values in South Australia. Fisher's exact test (2-sided) was used for categorical outcomes and Two-sample Wilcoxon rank-sum (Mann–Whitney) test for continuous outcomes [EQ-5D-5L index].

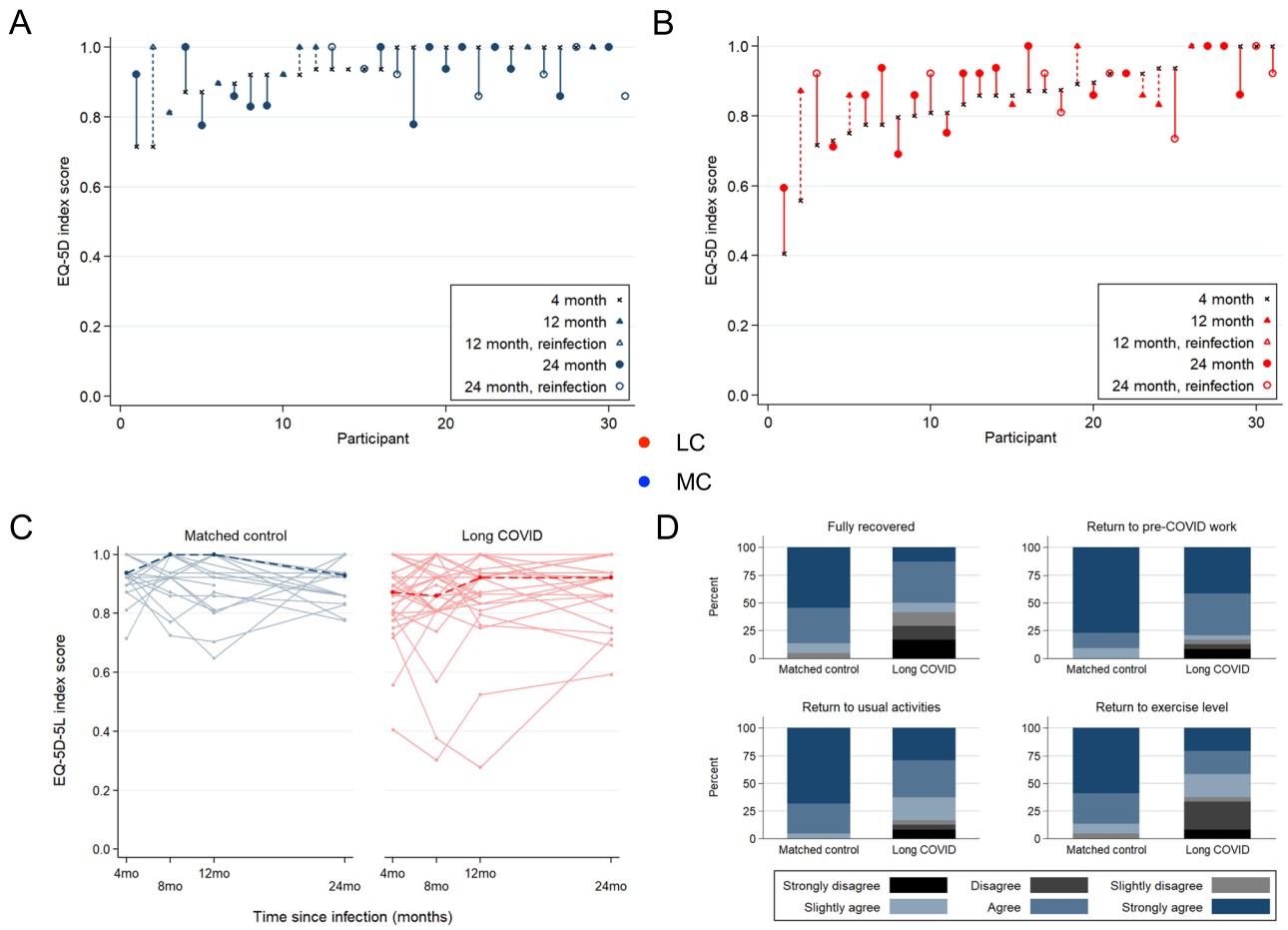

**Fig. 5 | Changes in health-related quality of life by EQ-5D-5L index score.** EQ-5D-5L index score at 4-month visit are ordered ascending on the x-axis for Matched Control (**A**) and Long COVID (**B**) participants. Vertical lines connect the participants' initial EQ-5D-5L score (4-month) and the last available EQ-5D-5L score (12- or 24-month). Crosses = EQ-5D-5L index score at 4-month visit, Triangle symbol = EQ-5D-5L index score at 12-month visit, Circle symbol = EQ-5D-5L index score at 24-month visit, Closed symbol = no COVID-19 reinfection during follow-up, Open symbol = COVID-19 reinfection during follow-up. **C** Median (dark dashed line) and participant (pale solid line) trajectories of EQ–5D-5L index score over study follow-up. **D** Participant-reported functional status at 24-months post-infection, related to full recovery from COVID, return to COVID-19 work, return to usual activities, and return to exercise level. LC Long COVID, MC matched controls.

effective anti-virals. In this cohort, most of whom were managed in the community, most aspects of immune dysregulation previously outlined through 8-months post-COVID among individuals with LC in the ADAPT cohort had recovered by two years following infection. This data is critically important in helping to define the natural history of LC over an extended follow-up period. By 24-months almost all

parameters which had shown striking differences between the LC and MC control groups at 4- and 8-months had resolved, with no significant differences remaining between the two groups. The exceptions to this were levels of IFNs β and γ, and spike- and NC-specific CD8[+] T cells, reasons for which are postulated below. Importantly, alongside the recovery in immune markers, we observed an overall improvement in

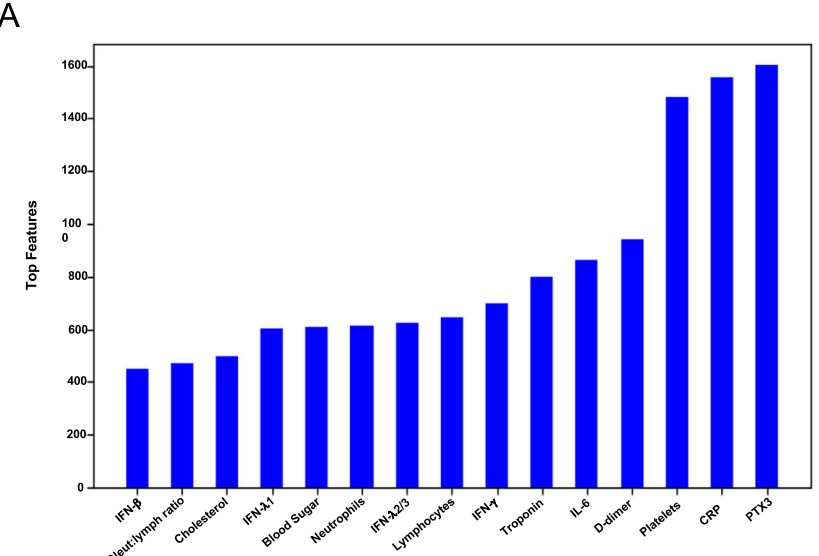

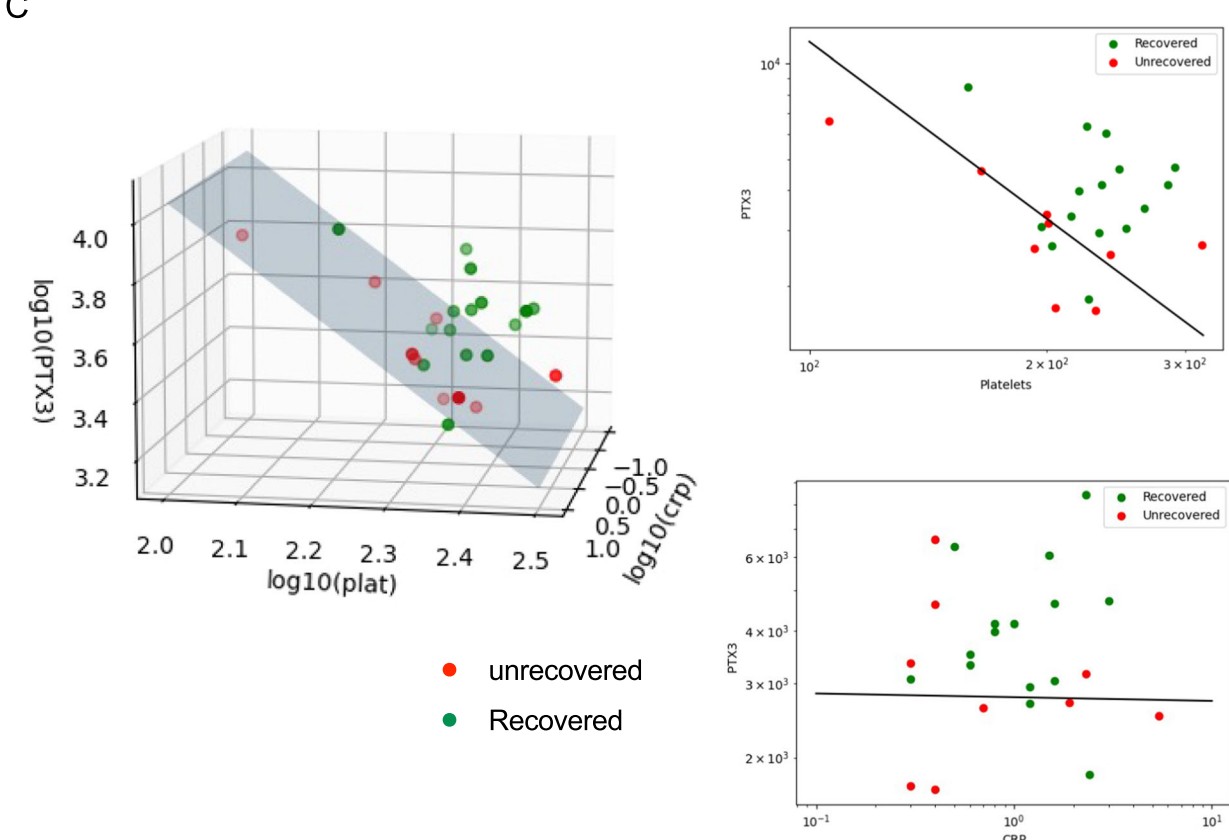

**Fig. 6 | Blood parameters associated with improvement in health-related quality of life at 24-months. A** Representative bar graph of log-linear model, showing frequency of features highly associated with recovery. **B** Table summarizing accuracy and F1 score for top 2 and top 3 most highly associated features. CI = 95% confidence interval. **C** Left-panel: 3-dimensional scatter plot of recovered vs unrecovered participant with concentration values of 3 markers (PTX3, CRP and platelets). Right-panel: 2D projections of PTX3 vs platelets (upper) and PTX3 vs. CRP (lower) with line representing the decision boundary. Recovered refers to improvement in health-related quality of life, unrecovered= no improvements, MC matched controls. Source data are provided as a Source Data file.

quality of life (QoL) in our LC participants. Whilst this was not universal it supports our immunological findings and a theory of overall slow return to health in most. The immunological and clinical reasons to explain the persistence of reduced QoL at 2 years in a minority of participants are also important to understand and will require further study.

We found that LC participants demonstrated higher neutralizing capacity at pre-vaccination timepoints. LC participants also exhibited elevated total nucleocapsid IgG levels compared to MC at 3-months. However, we found no difference in antibody levels at latter timepoints (12- to 24-months), which were potentially influenced by vaccination (spike) and re-infection (nucleocapsid and spike). These results were consistent with other studies that found elevated levels of spike IgG and IgA in LC at ~6–8-months[14,23] and studies that showed increased spike IgG and neutralization levels following vaccination[24,25]. Increased nucleocapsid IgG in both LC and MC at 24-months could be attributed to re-infection. A lack of discrepancy between B-cell frequencies in peripheral blood but higher antibody levels prior to vaccination suggests that in LC, these cells could be driven to produce antibodies via a pro-inflammatory milieu and/or persistent viral antigens in tissues.

Evaluation of T cell phenotype and function by assessing activation-induced markers and inhibitory receptors demonstrated that the frequencies of spike- and nucleocapsid-specific CD4$^+$ T cells were significantly higher in LC participants at 3- and 8-months, as also observed by ref. 25, but were returning towards levels seen in MC by 24-months; normalization of immune regulation and convalescence inclusive of recovery. Conversely spike- and nucleocapsid-specific CD8$^+$ T cells were consistently higher in LC participants at all timepoints. PD-1 and TIM-3 expression on both CD4$^+$ and CD8$^+$ T cell subsets were elevated in LC participants at 3- and 8-, but not at 24-months. This resolution of inhibitory markers was also seen when assessed at transcriptomic level by the exhaustion scores in CD4$^+$ and CD8$^+$ T cells at 24-months. Comparable levels of inhibitory receptors on T cells coincided with reduction of activated innate immune cells at 24-months and overall decrease in interferon expression. In addition, when single cell RNA-seq was performed, at 24-months proportions of immune cells in both innate and adaptive compartments were similar between LC participants and MC, including naïve T and B cell subsets which we had previously found to be relatively depleted in LC at 3- and 8-months[8]. These findings are consistent with normalization of immune dysregulation.

IRS at 24-months showed no significant differences between LC and MC in most innate cell subsets. While some ($n = 9$) LC participants had detectable IFN-β and IFN-γ, this was skewed by those with COVID-19 reinfection ($n = 6$) and significance was lost when reinfected participants were removed. Similarly, differentially expressed genes within CD14$^+$ monocytes contributed to a higher IRS in LC which could indicate their involvement in interferon expression due to activation following vaccination or re-infection[26,27]. Importantly, IRS levels were elevated in T cell subsets in LC compared to MC at early timepoints, and although these remained different between the groups, the extent of the difference decreased over time. Taken together, these findings suggest chronic stimulation of innate and adaptive arms of the immune system and the inflammation observed at earlier timepoints resolves over time.

Analysis of T cell clones revealed the presence of clones responsive to known antigens- in both groups, including SARS-CoV-2-specific T cells, with no noticeable differences in their exhaustion state, even though $n = 3$ of LC and $n = 1$ of MC were reinfected prior to sampling. Taken together, these findings demonstrate that T and B cells in LC do not express higher level of check point inhibition markers, or lower levels of naïve cells compared to MC, suggesting normalization of immune function over time.

Self-reported health-related quality of life analysis using the EQ-5D-5L score demonstrated that participants with LC reported more problems in mobility, usual activities, and pain/discomfort domains within the first few months after infection, but by 24 months these differences were less and no longer statistically significant. Initially, those with LC had lower overall EQ-5D index scores compared to MC, but there was no significant difference in median index scores at 24-months. Similarly, participants with LC were more likely to have "poor health" compared to MC at the initial visit, but this difference disappeared by the 24-month visit. Although these findings clearly show an overall trend to improvement over prolonged follow-up, it should be noted that these improvements were not seen in everyone and QoL did remain somewhat lower in the LC than MC, even though not statistically so. Additionally in terms of self-report, LC participants were less likely to agree with statements about recovery from COVID-19 and return to usual activities and exercise at the 24-month follow-up. The difference in these outcomes needs further exploration, including qualitative evaluation, but suggests that the lived experience and recovery of LC is likely to be complex and multifactorial. Irrespective of immunological recovery, other causes of poor health, including persisting organ damage, cognitive impairment[28], and the mental health impact of significant illness, may be contributing and mean that full physical recovery may lag behind immune recovery.

A log-linear classification model was used to analyze 15 blood parameters from LC participants at 24-months and found that PTX3, CRP, and platelet levels were highly associated with improvement of health-related quality of life. Pentraxins (PTX3 and CRP) are acute phase proteins synthesized when the body is stimulated by microbial invasion or tissue damage[29]. Decreased levels of these important pentraxins have been associated endothelial dysfunction in other studies[30,31] and dysregulated coagulation with reduced platelet levels has been was observed in other disease settings[32]. The finding that these biomarkers are associated with improvements in quality of life further supports a theory of gradual return to health underpinned by resolution of significant immunovascular dysregulation. However, this observation is likely to be of limited value in terms of day-to-day patient management as the changes seen are well within the normal 'healthy' range for both platelets and CRP (platelets 150–400 × 10$^9$/L; CRP < 5 mg/L). Further, and somewhat surprisingly, the LC who had recovered displayed levels of CRP, platelets and PTX3 that were marginally higher, but not statistically different from, than in the non-recovered LC. Plasma levels of these analytes were not significantly different, on either between group comparisons (Mann–Whitney) or across the three groups (Kruskal–Wallis) when compared across recovered LC, non-recovered LC and MC (Supplementary Fig. 5).

Our study has a few limitations, the sample size included are relatively small, especially for evaluating quality of life measures, and several participants were lost to follow-up over time. Nevertheless, these were relatively few (23% ($n = 7$) of LC and 26% ($n = 8$) of MC) and our cohort is unique in its ability to have repeated the same complex evaluations in the same individuals over 2-years of follow-up providing unrivalled data on return to health through a lens that combines both immunological and quality of life measures. Additionally, our definition of LC, initially set in mid 2020 and used in our prior analyses, is far narrower than subsequent accepted definitions. The inclusion of three of the commonest symptoms of LC however ensures that these findings are broadly relevant, as is the inclusion of predominantly community managed patients with mild illness. The use of patient-reported symptoms (rather than clinical measures) is in line with the International Consortium for Health Outcomes Measurement COVID-19 standard set[33], focusing primarily on outcomes that matter to patients. Our cohort mainly consists of patients who had acute infections of mild to moderate severity. While the majority have recovered, other studies based on patients followed after severe infection show that multiple symptoms may persist at 2-years[34]. However, it should be acknowledged that in these latter individuals,

who may have had significant end organ damage during the acute phase of their illness, there may be other drivers for their persistent symptomatology.

In summary, our data provides comprehensive evidence that the majority of measures of immunological dysfunction that we and others have previously reported up to 8-months in people with LC have resolved by 2-years in the majority (62%) of people with LC. Coupled with evidence of a general improvement in health-related quality of life measures from within the same individuals, this provides real optimism for people living with LC, and will be important for continuing to define the natural history of this new condition. Nevertheless, optimism must be tempered with caution and the understanding that in some individual's full health has not been recovered (38%) even 2-years post COVID-19, and research into the pathogenesis and prognosis of LC must continue. Our previous observations of immune abnormalities at earlier timepoints have now be confirmed in international studies[13,14,23,35]. However, it may be that the triggers of LC symptomatology are not the same as the factors that maintain them. That is the lack of symptomatic recovery in some patients despite immunological recovery may relate to an alternate underlying pathology as a cause for their LC symptoms, such as an element of persisting sub-clinical end organ damage or psychosocial trauma, or the triggering of other immunological process that we are not able to fully evaluate by sampling in the periphery.

## Methods
### Study design
ADAPT, is an ongoing prospective, observational cohort study of patients seen at St Vincent's Hospital Sydney (Australia) who tested PCR positive for SARS-CoV-2 infection, this cohort has previously been described in detail[8,36,37]. Each patient is followed for a period of 24-months from the time of diagnosis, with up-to 8 pre-specified timepoint collections. The study was approved by St. Vincent's Hospital, Sydney Human Research Ethics Committee (2020/ETH00964) and is a registered trial (ACTRN12620000554965). Gender was included in the original study design. Donors from both groups were age and gender matched. Median (IQR) age of groups was LC = $49.6 \pm 14.9$ and MC = $48.9 \pm 12.8$. Both groups contained 52% female and 48% male participants. Unexposed healthy donors were recruited through St Vincent's Hospital and was approved by St Vincent's Hospital, Sydney Human Research Ethics Committee (HREC/13/SVH/145). All participants provided written, informed consent before study procedures began.

### Clinical measures: long COVID classification
As previously described in the ADAPT cohort[36], participants were recruited during the initial wave of COVID-19 from hospital and community locations following mild/moderate severity of infection. Long COVID status was assigned if participants reported >1 persistent symptom of dyspnoea, chest pain, or fatigue/malaise at least 90 days after estimated date of initial infection. Age- and sex-matched participants who were asymptomatic were included in the current analysis as matched controls to participants with long COVID.

### Patient-reported outcomes
Comprehensive patient-reported outcome measures were assessed at all follow-up visits. Generic health-related quality of life was assessed using the EQ-5D-5L tool, which measures five dimensions of health (mobility, self-care, usual activities, pain/discomfort, and anxiety/depression) through a 5-level Likert scale (ranging from no problems to extreme problems). Health state utility values (range −0.25 to 1.00) were calculated for each participant visit based on their responses using the English value set[38] (version 1.1, updated 01/12/2020). To classify generic "poor health", participants index scores were compared to age- and sex-matched normative values from the general population of South Australia[39]. Participants were considered to have poor health if their EQ-5D-5L score was below than the lower 95% confidence interval of the population values. It is anticipated that the normative general population values in South Australia are comparable to the normative general population values where the participants were located (i.e., neighboring state of New South Wales).

To understand how COVID-19 specifically impacts daily activities, participants rated their agreement to four statements about functioning: (1) "I have fully recovered after COVID-19"; (2) "I feel confident returning to my pre-COVID work"; (3) "I have returned to my usual activities of daily living"; and (4) "I have returned to my normal exercise level". Responses were on a 6-level Likert scale (Strongly Disagree; Disagree; Slightly Disagree; Slightly Agree; Agree; Strongly Agree) and completed at the 24-month visit.

### Routine pathology
Hematology, biochemistry, and immune biomarkers (C-reactive protein, D-dimer, troponin I, total cholesterol, lymphocyte count, neutrophil count, and blood glucose) were assessed at each study visit as part of routine clinical care. Biomarkers were analysed using NATA accredited clinical chemistry/pathology platforms (SydPATH, St Vincent's Hospital, Sydney, Australia).

### Ex vivo phenotyping and combined CD4/CD8 T cell activation assay
Cryopreserved PBMCs were thawed using RPMI medium containing L-glutamine and 10% FCS (ThermoFisher Scientific, USA) supplemented with Penicillin/Streptomycin (Sigma-Aldrich, USW), and subsequently stained with monoclonal antibodies (mAb) binding to extracellular markers. Extracellular panel included: Live/Dead dye Near InfraRed, CD38 (HIT2, #MHCD3819 [1:100]) (ThermoFisher Scientific, USA); CD3 (UCHT1, # 300430 [1:100]), CD8 (HIL-72021, # 301042 [1:100]), CD123 (6H6, #306043 [1:50]), PD-1 (EH12.1, #562516 [1:100]), TIM-3 (TD3, #746771[1:100]), CD27 (L128, #563167 [1:100]), CD45RA (HI100, #564552 [1:100]), IgD (IA6-2, #561315 [1:100]), CD25 (2A3, #340939 [1:20]), and CD19 (HIB19, #557921 [1:100]) (BioLegend, USA); CD4 (OKT4, 300533 [1:100]), CD127 (A019D5, #351325 [1:100]), HLA-DR (L234, #307671 [1:100]), CCR7 (G043H7, 353217 [1:100]), CD16 (GB11, #302054 [1:100]), CD14 (HCD14, #325631 [1:100]), CD56 (NCAM-1, #557747 [1:100]), CD11c (B-ly6, #561355 [1:20]), and CD57 (QA17A04, #393304[1:100]) (BD Biosciences, USA). FACS staining of 48 h activated PBMCs was performed as described previously, but with the addition of CD137 (4B4-1) to the cultures at 24hrs. Final concentration of 10 µg/mL of SARS-CoV-2 peptide pools[24] (Genscript) were used and staphylococcal enterotoxin B (SEB; 1 µg/ml)(#S4881-MG, Sigma) was used as a positive control (ThermoFisher Scientific). In vitro activation mAb panel included: CD3 (UCHT1, # 300429 [1:100]), CD4 (RPA-T4, # 557922 [1:100]), CD8 (RPA-T8, # 301041 [1:100]), CD39 (A1, # 328205 [1:100]), CD69 (FN50, # 310911 [1:100]), CD137 (4B4-1, # 309820 [1:200]) all BioLegend, CD25 (2A3, #340939 [1:20]), CD134 (L106, # 340420 [1:10])- BD Biosciences. Samples were acquired on the Aurora CS spectral flow cytometer (Cytek Biosciences, USA) using the Spectroflo software v3.0 (Cytek). Prior to each run, all samples were fixed in 0.5% paraformaldehyde. Data analysis was performed using FlowJo version 10.7.1 (BD Biosciences).

### Flow cytometric detection of SARS-CoV-2 spike-reactive B cells
The Spike and Nucleocapsid gene of SARS-CoV-2 (isolate Wuhan Hu-1; NC_045512.2) was synthesized by GeneArt (Thermofisher) with a C-terminal polyhistidine tag, cloned into a standard CMV-driven expression plasmid, expressed in Expi293 cells (Thermofisher) and purified by Ni-NTA affinity and size-exclusion chromatography using a Superose 6 16/70 column (GE Healthcare). SARS-CoV-2 Spike was

biotinylated using BirA (Avidity). Biotinylated recombinant SARS-CoV-2 Spike was conjugated to streptavidin-BV421 (BD Biosciences). Recombinant SARS-CoV-2 Nucleocapsid was directly labeled to APC or PE using the Lightning-Link Kit (Abcam). PBMCs were thawed and stained with Aqua viability dye (Thermo Fisher Scientific) and then surface stained with Spike probes, CD19-ECD (J3-119, #IM2708U [1:150]) (Beckman Coulter), IgD AF488 (polyclonal) (#2030-30 [1:75]) (Southern Biotech), IgG-BV786 (G18-145, #564230 [1:75]), CD21-BUV737 (B-ly4, #564437 [1:150]), CD38 AF700 (HIT2 #560676 [1:100]), Streptavidin BV510 (#563261 [1:600]) (BD Biosciences), CD14-BV510 (M5E2, #301841), CD3 BV510 (OKT3, #317332, [1:600]), CD8a-BV510 (RPA-T8, #301048 [1:600]), CD16-BV510 (3G8, #302048 [1:500]), CD10-BV510 (HI10a, #312220 [1:750]), CD20 APC-Cy7 (2H7, #302314 [1:150]), CD27-BV605 (O323, #302829 [1:150]),CD71 PE Cy7 (CY1G4 #334112 [1:100])(BioLegend). Cells were washed twice with PBS containing 1% FCS and fixed with 1% formaldehyde (Polysciences) and acquired on a BD LSR Fortessa using BD FACS Diva.

## Anti-spike and anti-nucleocapsid diagnostic serology
Antibodies to SARS-CoV-2 spike in serum samples from ADAPT participants were measured using using the Euroimmun diagnostic ELISA for IgG anti-S1 (Luebeck, Germany). Antibodies to SARS-CoV-2 Nucleocapsid were measured using Euroimmun NCP diagnostic ELISA assay (Euroimmun). All assays were done according to the manufacturers' instructions.

## Live virus neutralization assay
Rapid high-content neutralization assay with HAT-24 cells was done as previously described[40]. Briefly, human sera were serially diluted (1:2 series starting at 1:10) in DMEM-5% FBS and mixed in duplicate with an equal volume of SARS-CoV-2 virus solution standardized at $2\times$ $VE_{50}$[41]. After 1 h of virus–serum coincubation at 37 °C, 40 µL were added to an equal volume of nuclear-stained HAT-24 cells pre-plated in 384-well plates as above. Plates were incubated for 20 h before enumerating nuclear counts with a high-content fluorescence microscopy system as indicated above. The % neutralization was calculated with the formula: $\%N = (D - (1 - Q)) \times 100/D$ as previously described[1]. Briefly, "Q" is a well's nuclei count divided by the average count for uninfected controls (defined as having 100% neutralization) and $D = 1 - Q$ for the average count of positive infection controls (defined as having 0% neutralization). Sigmoidal dose–response curves and $IC_{50}$ values (reciprocal dilution at which 50% neutralization is achieved) were obtained with GraphPad Prism software.

## Serum analytes
The LEGENDplex custom-made panel (IL-6, IFN-β, IFN-λ1, IFN-λ2/3, IFN-γ and PTX3) were purchased from BioLegend, and assays were performed as per the manufacturer's instructions. Beads were acquired and analyzed on a BD Fortessa X20 SORP (BD Biosciences). Samples were run in duplicate, and 4000 beads were acquired per sample. Data analysis was performed using Qognit LEGENDplex software (BioLegend). Lower limit of detection values was used for all analytes at the lower limit.

## Log-linear Model
The analytes most associated with long COVID were identified via Log-Linear Classification. For an arbitrary set of 3 analytes, let the concentration of the ith analyte at 24 months be denoted $w_i$. Log-Linear Classification assigns a weight $a_i$ to the logarithm of each analyte concentration. A linear function of these logged concentrations and weights takes the form $f(\vec{a})$ is a threshold parameter. The weights $w_i$ as well as the intercept $w_0$ are selected to maximize the predictive power of the linear classifier by training on the analyte data, where $f(\vec{a}) > 0$, results in the classifier predicting that the participant with analyte concentration $\vec{a}$ has long COVID and does not have

long COVID otherwise.

$$f(\vec{a}) = w_0 + \sum_{i=1}^{N} w_i \log_{10}(a_i) \qquad (1)$$

Due to the modest small sample size of 24 participants at month 24, we performed bootstrapping to randomly sample new populations of size 24 from our population with replacement. Then the sampled population was split 15:9 into test and train datasets. The training dataset was used to train a log-linear classifier using Python3 v3.8.10 and the Scikit-learn machine learning package v0.24.1. From the test set, the number of true positives (TP: both the classifier and data indicate the participant had long COVID), true negatives (TN: both the classifier and data indicate the participant had asymptomatic COVID), false positives (FP: classifier predicts the participant will have long COVID, but the data disagrees) and false negatives (FN: classifier predicts the participant will have asymptomatic COVID, but the data disagrees) were identified. Then two subsequent scores are calculated. Firstly, the accuracy is defined as $(TP + TN)/(TP + TN + FP + FN)$ and measures the proportion of test participants that had their COVID status correctly predicted. The second measure is the F1 score and is defined as $TP/(TP + 0.5*(FP + FN))$, which is a measure that combines recall, how many long COVID cases were correctly predicted, and precision, of all the participants predicted to have long COVID how many were correct. This process is repeated for 2000 different bootstrapped sample populations. The average accuracy of a model of N analytes is then calculated and used to assess which combination of N analytes performs the best.

## Single cell RNA-seq analysis PBMCs and sequence alignment
Briefly, PBMC from 20 individuals from ADAPT cohort (10 LC and 10 MC) at 4-, 8- and 24-month timepoints were genotyped and had PBMCs sequenced using the 10x genomics platform. For each individual, 1 vial (~1 mL) of biobanked (frozen) PBMCs was retrieved and thawed. Cell density per vial was roughly $7 \times 10^6$ cells/mL. To this end ~30 µL was used for single-cell RNA-sequencing, and another fraction ($10^6$) for genotyping. Cell viability was tested using trypan blue, with high levels ~>90% for all samples. Live cells were pooled (multiplexed) with 10 participant samples per pool in replicate, for a total of 4 batches. For each batch, single-cell RNA capture and barcoding with the Single Cell 5' v2.0 NextGEM single cell RNA seq (GEX (#PN-1000165) + VDJ (TCR + BCR) (#PN-1000005)) from 10x Genomics, with a target capture of 20,000 cells was performed. Sequencing was done with the Illumina NovaSeq 2000 on a S4 flowcell. We aimed for 30,000 Reads Per Cell (RPC) for GEX, and 7500 RPC for both BCR and TCR respectively. Reads were processed and demultiplexed the using Cell Ranger Single Cell Software Suite (v 7.0.0; 10x Genomics). Mapping and alignment were done to GRCh38 (GEX: refdata-cellranger-GRCh38-2020-A, and VDJ: refdata-cellranger-vdj-GRCh38-alts-ensembl-5.0.0) using STAR within the Cell Ranger Suite. The pipeline was executed on a high-performance cluster with a 3.10.0-1160.42.2.el7.x86_64 operating system.

## Genotyping and demultiplexing
DNA was extracted using Qiagen QIAamp DNA mini kit and genotyping was done using the UKB Axiom array. The genotypes were called using the Axiom Analysis Suite from Thermofisher (AxAS v5.1) following the Best Practices Genotyping Analysis Workflow in the Axiom_UKB_WCSG.r5 library. Imputation was performed using the Michigan Imputation Server with Eagle (v2.3) for phasing, and Minimac4 (v1.0.2) for imputation and the 1000 Genomes project reference panel (1000 G 30X WGS reference panel). Missingness was assessed with vcftools (v0.1.16). Two samples (AD007 and AD322) were rerun with adjusted parameters due to high missingness (>5%). After imputation,

we filtered on MAF > 0.05 and R2 > 0.3 using bcftools (v1.10.2.). Demultiplexing was performed to assign cells to individuals using Demuxlet.

### Quality control and cell classification

Each pool was processed individually, with cells assigned as doublets excluded from the analysis, along with cells with higher than 5% mitochondrial gene expression. Following the quality control processing, Seurat was used to integrate the individual pools after normalization using the SCTransform function. To cluster and classify cells, VDJ genes were removed from the integrated object counts by matching gene names to immunoglobulins (IG) with: "IG[HLK][VJ]" or "IGHD-", and T-cell receptors (TR): "TR[ABGD][VDJ]". SCTransform was then used on "non-VDJ" count data to normalize the expression levels. Cells were then classified using Azimuth/Seurat pipeline with the human PBMC reference for L2 and the metadata was then added to the full count expression data. All further analyses were performed using the integrated object with the full expression data in R Statistical Software (v4.1.3; R Core Team 2022).

### Single-cell analysis and cell scores

With the full object, cell-type proportions were calculated for individuals split by long-COVID status and tested for significance using the propeller function in the speckle R package. Using the AddModuleScore function in Seurat, a CD8 exhaustion score and an interferon score (IFN) were calculated, along with T stem cell memory and T cytotoxic scores, based on gene sets from ref. 20.

### Post-processing of 10x VDJ datasets

Filtered VDJ contigs generated by Cell Ranger were post-processed with IgBLAST (version 1.1.9) to align against the IMGT human Reference Directories to generate AIRR-C tab-delimited output. For cells with multiple chains for the same loci the chain with the highest UMI count was retained. T and B cell VDJs were filtered to remove non-productive chains (stop codons or out-of-frame) and chains that lacked CDR3's and analysis was restricted to cells that were Singlets.

To defined B cell clonal lineages, CDR3s from B cell IGH/K/L were extracted and CDR3 nucleotide sequences were binned based on V gene, J gene and CDR3 length for clustering with cd-hit (version 4.7) using the cd-hit-est tool. Clustering was undertaken at a 90% identity threshold for cells from each individial and B cells that shared the same cluster membership for IGH and IGK/L were considered clonally related. For T cells, clonotypes were defined by shared V gene, J gene and CDR3 amino acid sequence.

To annotate putative antigen specificity for T cells two databases of TCRs of known specificity were obtained; immuneCODE for SARS-CoV-2 specific TCRs and VDjdb (version 2022-02-30) for SARS-CoV-2 plus other antigens reported in literature. TCR clonotype putative specificity were annotated by exact TRB clonotype matches (same V, J, and CDR3 AA sequence for TCR beta loci). Where the same clonotype was associated with more than one antigen the TCR specificity was flagged as 'multiple'. For the B cells, to explore SARS-CoV-2 specificity, sequences were obtained from CoV-AbDab version 210223. CDR3 AA sequences from the IGH of known specificity were clustered with CDR3s from the 10x datasets using cd-hit at an 80% identity threshold. If an IGH of known specificity clustered with IGH from 10x then those cells were annotated with the specificity from the CoV-AbDab database. VDJ data from Cell Ranger and IgBLAST were integrated with GEX analysis and putative antigen specificity in R (version 4.3.0) within RStudio IDE (version 2023.3.1.446) using the tidyverse package (version 2.0.0).

### Statistical analysis

All column graphs are presented as medians with inter-quartile ranges. For unpaired samples Mann–Whitney U test was used employing Prism 10 (GraphicPad, La Jolla, CA, USA) software. $p < 0.05$ were considered significant (*<0.05, **<0.01 and ***<0.001). Analysis of patient reported outcome measures was conducted using Stata v14 (StataCorp LLC, College Station, TX, USA).

### Reporting summary

Further information on research design is available in the Nature Portfolio Reporting Summary linked to this article.

## Data availability

Clinical data from this study are held by the data management team, and can be made available to other research groups, after approval of a proposal by ADAPT steering committee. To protect patient privacy, underlying electronic health records may be accessed via a remote server pending data access agreement. Please contact the corresponding author with an outline of the intended use. scRNAseq data generated in this study have been deposited in a public database under accession code SRA: SRP497951, Bio Project: PRJNA1092125 and GEO: GSE262861. All other data are available in the article and its Supplementary files or from the corresponding author upon request. Source data are provided with this paper.

## Code availability

Codes used for scRNAseq analysis have been deposited in the Zenodo database [https://zenodo.org/records/10888516]. Access to codes can be found here https://doi.org/10.5281/zenodo.10888516.

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

## Acknowledgements

We thank the staff at the St Vincent's Centre for Applied Medical Research–Clinical Trials Unit for their expertise in specimen processing and biobanking. We thank E. Johansson Beves for assistance with the Cytek Aurora. We appreciate grant support from the St Vincent's Clinic Foundation, the Curran Foundation, the Rapid Response Research Fund, and the Medical Research Futures Fund (Australia). D.B.W. was supported by NSF-DMA grant 1902854. S.J.K. was supported by the Victorian Government, MRFF award 2005544, NHMRC program grant 1149990 (S.J.K. and A.D.K.) and NHMRC fellowships (1136322 to S.J.K.). S.B. was supported by a fellowship from the Magid's and a Ramsay award by the Solve ME/CFS Initiative (SMCI).

## Author contributions

Protocol design and clinical management: G.V.M., G.J.D., D.R.D., A.D.K., P.C., A.B., B.J.B. Experimental design and procedures: C.P., B.M., V.K., A.Ag., A.Ak., V.M., M.S., P.C., H.X.T., A.W., S.G.T., S.J.K. Data analysis and Visualization: C.P., B.J., K.L.J., S.B., D.B.W. Writing - original draft: C.P., B.J., A.D.K., G.V.M. Writing - review & editing: C.P., B.J., A.D.K., G.V.M., G.J.D., B.M., V.K., K.L.J., S.B., D.B.W. All authors contributed to the manuscript and approved the submitted version.

## Competing interests

G.J.D. received grants from Gilead, Abbvie, Merck and Bristol-Myers Squibb, personal fees from Gilead, Abbvie and Merck, and nonfinancial support from Gilead, Abbvie and Merck, all outside the scope of the submitted work. B.J.B. received grants from St Vincent's Clinic during the conduct of the study; and consults for AbbVie, Janssen and Viiv, and grants from Biogen, all outside the scope of the submitted work. All other authors declare that there are no competing interests.
