## [Peer Review File · Nature Communications]

Improvement of immune dysregulation in individuals with long COVID at 24-months following SARS-CoV-2 infectionEditorial Note: This manuscript has been previously reviewed at another journal that is not operating a transparent peer review scheme. This document only contains reviewer comments and rebuttal letters for versions considered at *Nature Communications*.

REVIEWER COMMENTS

Reviewer #1 (Remarks to the Author):

The authors have largely addressed my previous comments.

Reviewer #2 (Remarks to the Author):

In your manuscript by Phetsouphanh et. al., you evaluate ADAPT cohort participants at later time points than in your previous publication describing the clinical and immunologic features of this Long COVID (LC) cohort. As noted in my last review, the techniques used are state of the art, including live virus neutralization. LC participants showed elevated spike and nucleocapsid IgG levels, higher neutralizing capacity, and increased spike- and nucleocapsid-specific CD4+ T cells, PD-1, and TIM-3 expression on CD4+ and CD8+ T cells at 3- and 8-months, but these differences did not persist at 24-months. Single-cell RNA sequencing at 24-month timepoint revealed similar immune cell proportions and reconstitution of naïve T and B cell subsets in LC. No significant differences in exhaustion scores or antigen-specific T cell clones were observed. These findings suggest resolution of immune activation in LC and return to comparable immune responses between LC and age and sex matched controls over time. The manuscript is greatly improved by the addition of RNAseq data at earlier time points, further supporting that there are abnormalities in LC that resolve over time. However, there are no specific immune cell differences that persist in the 38% of those who note no improvement in quality of life, making it unclear if the immune abnormalities at 4 and 8 months are specific to the symptoms of LC. This comment was not addressed except in noting that other factors other than immune abnormalities may be involved. In addition, the rebuttal contains a response to my comment about stating there is a relationship between LC and CRP, PTX3 and platelets but the manuscript itself does not make clear that the unrecovered LC participants have lower levels of CRP, PTX3

and platelets compared to recovered. It continues to note that “PTX3, CRP and platelet levels were highly associated with improvement of health-related quality of life”. It does now state that the levels for both groups were in the normal range with important caveats to their use clinically noted, which is an improvement. You should note clearly that levels are lower in unrecovered LC participants, which does not make complete sense with CRP. References to prior work in this field are appropriate. My prior concerns on other points have been addressed.

REVIEWER COMMENTS

Reviewer #1 (Remarks to the Author):

The authors have largely addressed my previous comments.

Reviewer #2 (Remarks to the Author):

In your manuscript by Phetsouphanh et. al., you evaluate ADAPT cohort participants at later time points than in your previous publication describing the clinical and immunologic features of this Long COVID (LC) cohort. As noted in my last review, the techniques used are state of the art, including live virus neutralization. LC participants showed elevated spike and nucleocapsid IgG levels, higher neutralizing capacity, and increased spike- and nucleocapsid-specific CD4+ T cells, PD-1, and TIM-3 expression on CD4+ and CD8+ T cells at 3- and 8-months, but these differences did not persist at 24-months. Single-cell RNA sequencing at 24-month timepoint revealed similar immune cell proportions and reconstitution of naïve T and B cell subsets in LC. No significant differences in exhaustion scores or antigen-specific T cell clones were observed. These findings suggest resolution of immune activation in LC and return to comparable immune responses between LC and age and sex matched controls over time. The manuscript is greatly improved by the addition of RNAseq data at earlier time points, further supporting that there are abnormalities in LC that resolve over time. However, there are no specific immune cell differences that persist in the 38% of those who note no improvement in quality of life, making it unclear if the immune abnormalities at 4 and 8 months are specific to the symptoms of LC. This comment was not addressed except in noting that other factors other than immune abnormalities may be involved. In addition, the rebuttal contains a response to my comment about stating there is a relationship between LC and CRP, PTX3 and platelets but the manuscript itself does not make clear that the unrecovered LC participants have lower levels of CRP, PTX3 and platelets compared to recovered. It continues to note that “PTX3, CRP and platelet levels were highly associated with improvement of health-related quality of life”. It does now state that the levels for both groups were in the normal range with important caveats to their use clinically noted, which is an improvement. You should note clearly that levels are lower in unrecovered LC participants, which does not make complete sense with CRP. References to prior work in this field are appropriate. My prior concerns on other points have been addressed.

Our Response-

We thank the reviewers for their critique of the revised manuscript. We have addressed the constructive concerns of reviewer 2 by the following revisions to the manuscript. All changes are highlighted in the manuscript.

- 1) We also acknowledge that there is a less than clear association between the immune profile and LC recovery or lack thereof. The immune abnormalities we report at earlier timepoints have also been observed in other international studies; San Cruz *et al* also found elevated IFN-beta at 6-months in their PASC cohort, Augustin *et al* observed highly activated plasmacytoid dendritic cells, Klein *et al* detected high levels of HLA-DR+ on pDC and PD1+Tim-3+ expression on T cells, and more recently Yin et al

showed elevated levels of SARS-CoV-2 antibodies and more exhausted T cells in Long COVID at 8-months post-infection. These international cohort studies confirm our previous observation of immune dysregulation at 4 and 8-months. Our current study is the first to show the immune profile of long COVID at the 2-years and the first to describe any associations with recovery. We have therefore added the following sentence to the last paragraph of the discussion at lines 419-425.

Our previous observations of immune abnormalities at earlier timepoints have now been confirmed in several international studies^{1,2,3,4}. However, it may be that the triggers of LC symptomatology are not the same as the factors that maintain them. That is the lack of symptomatic recovery in some patients despite immunological recovery may relate an alternate underlying pathology as a cause for their LC symptoms, such as an element of persisting sub-clinical end organ damage or psychosocial trauma, or the triggering of other immunological process that we are not able to fully evaluate by sampling in the periphery.

References:

1. Augustin, M., *et al.* Immunological fingerprint in coronavirus disease-19 convalescents with and without post-COVID syndrome. *Front Med (Lausanne)* **10**, 1129288 (2023).
2. Klein, J., *et al.* Distinguishing features of long COVID identified through immune profiling. *Nature* **623**, 139-148 (2023).
3. Santa Cruz, A., *et al.* Post-acute sequelae of COVID-19 is characterized by diminished peripheral CD8(+)beta7 integrin(+) T cells and anti-SARS-CoV-2 IgA response. *Nat Commun* **14**, 1772 (2023).
4. Yin, K., *et al.* Long COVID manifests with T cell dysregulation, inflammation and an uncoordinated adaptive immune response to SARS-CoV-2. *Nat Immunol* **25**, 218-225 (2024).

- 2) With regards to the levels of the differentiating biomarkers: CRP, Platelets and pentraxin 3 we have added the following sentences to the discussion:

The finding that these biomarkers are associated with improvements in quality of life further supports a theory of gradual return to health underpinned by resolution of significant immunovascular dysregulation. However, this observation is likely to be of limited value in terms of day-to-day patient management as the changes seen are well within the normal 'healthy' range for both platelets and CRP (platelets 150-400 x10⁹/L; CRP < 5mg/L). Further, and somewhat surprisingly, the LC who had recovered displayed levels of CRP, platelets and PTX3 that were marginally higher, but not statistically different from, than in the non-recovered LC. Plasma levels of these analytes were not significantly different, on either between group comparisons (Mann-Whitney) or across the three groups (Kruskal-Wallis) when compared across recovered LC, non-recovered LC and MC (Extended Data 5).

This has now been added to lines 384-393 in the discussion section and additional figure added as extended data 5 (see below).

Extended Data 5

Extended Data 5. Blood biomarkers at 24-months. A) Dot plots showing concentrations of CRP, Platelets and PTX 3 at 24-months; recovered (green), un-recovered (red), matched controls (blue).

REVIEWERS' COMMENTS

Reviewer #2 (Remarks to the Author):

I do not doubt the validity of the findings in the study as consequences of COVID and agree that others have observed similar findings. Whether they cause the symptoms of long COVID is not clear based on the fact that the immunologic abnormalities resolve even when symptoms do not. My concerns about the link between long COVID symptoms and the immunologic findings have been mitigated by the language in the revised manuscript and all my concerns have been addressed.